# Generation of Antibody-Drug Conjugate Resistant Models

**DOI:** 10.3390/cancers13184631

**Published:** 2021-09-15

**Authors:** Lucía Gandullo-Sánchez, Alberto Ocaña, Atanasio Pandiella

**Affiliations:** 1Instituto de Biología Molecular y Celular del Cáncer, CSIC, IBSAL and CIBERONC, 37007 Salamanca, Spain; lgandullo@usal.es; 2Hospital Clínico San Carlos, 28040 Madrid, Spain; albertocana@yahoo.es; 3Symphogen, DK-2750 Ballerup, Denmark

**Keywords:** ADCs, drug resistance, preclinical models, therapy

## Abstract

**Simple Summary:**

Antibody-drug conjugates (ADCs) constitute new and effective therapies in cancer. However, resistance is frequently observed in treated patients after a given period of time. That resistance may be present from the beginning of the treatment (primary or de novo resistance) or raise after an initial response to the ADC (secondary resistance). Knowing the causes of those resistances is a necessity in the field as it may help in designing strategies to overcome them. Because of that, it is necessary to develop models that allow the identification of mechanisms of resistance. In this review, we present different approaches that have been used to model ADC resistance in the preclinical setting, and that include the use of established cell lines, patient-derived ex vivo cultures and xenografts primarily or secondarily resistant to the ADC.

**Abstract:**

In the last 20 years, antibody-drug conjugates (ADCs) have been incorporated into the oncology clinic as treatments for several types of cancer. So far, the Food and Drug Administration (FDA) has approved 11 ADCs and other ADCs are in the late stages of clinical development. Despite the efficacy of this type of drug, the tumors of some patients may result in resistance to ADCs. Due to this, it is essential not only to comprehend resistance mechanisms but also to develop strategies to overcome resistance to ADCs. To reach these goals, the generation and use of preclinical models to study those mechanisms of resistance are critical. Some cells or patient tumors may result in primary resistance to the action of an ADC, even if they express the antigen against which the ADC is directed. Isolated primary tumoral cells, cell lines, or patient explants (patient-derived xenografts) with these characteristics can be used to study primary resistance. The most common method to generate models of secondary resistance is to treat cancer cell lines or tumors with an ADC. Two strategies, either continuous treatment with the ADC or intermittent treatment, have successfully been used to develop those resistance models.

## 1. Introduction: ADC Structure and Mechanism of Action

ADCs are compounds that combine the specificity of a monoclonal antibody (mAb) with the toxicity of a cytotoxic payload (Figure 1).

The goal of these molecules is to transport the cytotoxic agent to tumor cells, maximizing efficacy and minimizing toxicity against non-tumoral tissues [1,2]. An ADC is made of three components: (i) the mAb, (ii) a linker, and (iii) the cytotoxic drug (Figure 1) [3]. The mAb must recognize a tumor-associated antigen. That antigen must be located at the cell surface. Desirably, but not mandatory, the antigen must be overexpressed in the cancer cells, a circumstance that facilitates the targeting of the ADC to the tumor with preference to normal tissues also expressing the membrane target protein. Another requirement is that the targeted molecule, mainly a protein, must be internalized together with the compound to deliver the cytotoxic agent inside tumor cells [4].

A second component of the ADC is the linker, a chemical structure that is used to bind the mAb to the cytotoxic agent. The linker must be stable in circulation to prevent unspecific drug release and therefore possible damage to normal cells [5]. The conjugation process must avoid altering the physicochemical properties of the mAb to guarantee the same binding affinity and specificity.

Two main classes of linkers are currently used in the ADC field: cleavable and non-cleavable (Figure 2). Cleavable linkers release the cytotoxic agent when they are processed under the presence of a chemical condition (such as low pH or reducing environment) or specific enzymes present in different cellular compartments. On the other hand, non-cleavable linkers require proteolytic degradation of the ADC to release the active cytotoxic [6]. Such a cleavage step is expected to occur in lysosomes, where ADC-target protein complexes are directed upon interaction from the cell surface. Their arrival to those organelles exposes them to the action of acidic lysosomal proteases that cleave the ADC and generate small peptide fragments. Some of those fragments contain the cytotoxic compound and are sufficiently small enough to be transported to the cytosol with the help of lysosomal membrane transporters [1,7,8].

The cytotoxic payloads used for the construction of ADCs include microtubule-disrupting agents (auristatin analogs, maytansinoids, and tubulysins), and DNA-damaging agents (calicheamicins, duocarmycins, and pyrrolo-benzodiazepines) (Figure 2). The first class induces mitotic arrest, while the second induces DNA damage. Both mechanisms finally cause cell death [9,10,11]. Considering moxetumomab pasudotox as an ADC, a third group of payloads, acting on protein synthesis, can also be included. This latter ADC includes PE38, a 38 kDa fragment of Pseudomonas exotoxin A that acts as a potent inhibitor of protein synthesis [12].

## 2. Approved ADCs

At present, eleven ADCs (Table 1) have been approved by the FDA, and many others are under clinical evaluation. In the last years, significant efforts have been done in the ADC field to engineer different versions of these drugs. Efforts have focused on the generation of multivalent and multispecific antibodies, site-specific conjugations, different linkers, and new effective payloads. The final aim of these different modifications is to improve ADCs therapeutic effectiveness augmenting the binding and reducing the effect on non-transformed cells [13,14].

## 3. Preclinical Models of ADC Resistance

The plasticity of tumoral cells allows them to adapt and survive to antitumoral drugs. Resistance to these drugs can be primary (also termed de novo or intrinsic) or secondary. Primary resistance occurs when there is no response from the beginning of the treatment because tumor cells have some inherent molecular features that prevent them from responding to the antitumoral drug. On the other hand, secondary resistance refers to tumors that have an initial response to the drug but develop resistance over time [15].

Two types of secondary resistance have been described. Adaptive resistance occurs when a clone or several clones of resistant cells are present in a heterogeneous tumor. The resistant clones usually represent a minority with respect to sensitive populations. Treatment with the antitumoral drug reduces tumor volume by acting on the largest sensitive population but is not as effective against resistant clones, that can survive via feedback mechanisms. That circumstance allows resistant cells to grow and become predominant in the relapsed tumor. This type of secondary resistance is considered the most frequent [16].

The other type of secondary resistance is termed acquired resistance, characterized by the appearance of novel molecular alterations in tumoral cells while being treated with the antitumoral drug. This type of resistance can be found under treatments that are highly mutagenic. For example, it has been reported that glioblastoma multiforme patients being treated with temozolomide, which is a mutagenic drug, present molecular alterations in their tumors that were not present at the initiation of the therapy [17,18].

### 3.1. In Vitro and Ex Vivo Models of Resistance to ADCs

Preclinical models of resistance to ADCs have mainly been generated using human cultured cell lines (Figure 3).

Several reasons are responsible for the choice of those preclinical models, including the availability of unlimited amounts of cells, good cost/effectiveness relation, uniformity of the cellular models, easy handling and storage, as well as avoidance of ethical objections associated with the use of animals or human tissues. On the other side, immortalized cultured cell lines have some limitations, including altered genomic content or the lack of stromal components of the tumor, that may contribute to the resistant phenotype [19]. The problem of genomic alterations that raise along with prolonged in vitro maintenance of tumoral cells could be bypassed by the use of primary cells, which are expected to retain genetic alterations and physiological characteristics closer to the clinical situation. However, primary tumoral cells are more difficult to obtain and their proliferation properties are usually more limited [20].

#### 3.1.1. Primary Resistance to ADCs in Established Cell Lines

Increased expression and/or activity of drug extrusion pumps have been widely studied as mechanisms of chemotherapy resistance [21,22]. The expulsion pumps belong to the ATP binding cassette (ABC) transporter family and cause the development of multidrug resistance (MDR). MDR is a type of resistance to multiple and structurally unrelated compounds. For example, cytotoxic agents of ADCs such as maytansinoids, are substrates for some of these transporters. Specifically, it has been reported that alterations in the expression of ABCB1/MDR1/P-gp, ABCC1/MRP1, ABCC2, and ABCG2/BCRP/MXR/ABCP lead to resistance to trastuzumab-DM1 (T-DM1) in preclinical models [23,24,25,26,27]. In these studies, inhibition of transporter activity restored sensitivity to T-DM1. These pumps have also been linked to resistance to other ADCs such as gemtuzumab ozogamicin (GO) and brentuximab vedotin (BV) [28,29,30,31,32,33,34].

High drug extrusion activity has been linked to resistance to ADCs in different cell lines. For example, the KG-1 acute myelogenous leukemia (AML) cell line and its variant subline KG-1a have been reported to be primary resistant to GO, even though they express CD33, the target of the antibody [35,36,37]. Such a resistant phenotype could be at least in part due to high pump activity [36]. Walter et al. reported that the human erythroblastic TF1 cell line was resistant to GO. They also reported that two sublines of HL60 resistant to anthracyclines (HL-60/AR) and vincristine (HL-60/VCR) were also resistant to GO [29,30]. GO resistance in TF1 could be partially explained by high MRP activity and HL60/AR and HL60/VCR were reported to present overexpression of ABCC1/MRP1 or ABCB1/MDR1/P-gp, respectively. Rosen et al. also confirmed primary resistance to GO in the TF1 cell line [37].

JIMT-1 is a HER2+ breast cancer cell line with primary resistance to the anti-HER2 antibody trastuzumab [38]. In addition, several scientists have reported that JIMT-1 is also resistant to the trastuzumab-derived ADC T-DM1, mainly in JIMT-1 xenograft models [23,39,40,41,42,43,44]. Recently, Yamazaki et al. generated a derivate JIMT-1 cell line that overexpressed ABCB1/MDR1/P-gp and reported resistance to T-DM1 [27]. In addition, other authors treated JIMT-1 with T-DM1 to increase the resistant phenotype [25,45]. Another example of primary resistance is the cell line SNU-216. SNU-216 is a HER2+ trastuzumab-resistant gastric cancer cell line that is also resistant to T-DM1 [7,23,46]. The HR6 cell line, derived from a trastuzumab-resistant BT474 xenograft, is also resistant to T-DM1 in vitro and in vivo [47,48].

In summary, the above-commented studies point to overexpression of ABC transporters as an important mechanism of primary resistance to ADCs.

#### 3.1.2. Secondary Resistance to ADCs in Established Cell Lines

The development of cancer cell lines resistant to chemotherapeutics has been a useful approach to investigate the mechanisms of cytotoxicity and resistance to those drugs. Indeed, McDermott et al. discussed the methodology for the in vitro development of drug-resistant cancer cell lines. Development of drug-resistant cell lines can be made with intermittent (pulsed) or chronic (or continuous) exposure [49]. Both approaches have also been used for the generation of ADC-resistant cell lines. Table 2 summarizes some examples of the methodology used to generate resistant cell lines by both approaches as well as the stability of the resistant phenotype in vivo. The table also summarizes the uncovered causes of the resistance present in those models, even though the present review is not focused on that. A detailed description of the causes of resistance identified by using those models has been published elsewhere [1,7].

Commonly, continuous treatment is performed using several doses of the ADC [50]. These doses are selected on the basis of the potency of the antibody found while assessing in vitro efficacy in cell proliferation/cytotoxic assays. Usually, the doses selected include a maximal dose (generally in the 5–10 nM range) as well as suboptimal doses (10 pM–2.5 nM). At the selected doses, the cell lines are followed visually along time to inspect for cell death. In the case of adherent cells, that is manifested by cell rounding and loss of brightness in phase-contrast microscopy. After this initial phase of selection of drug-resistant cells, an increase in the dose can be considered. Thus, for example, cells growing in 2.5 nM of an ADC can be shifted to culture media containing 5–10 nM of that ADC. That additional pressure may result in the loss of several clones resistant to low doses of the ADC, but on the other hand, it is expected to favor the generation of clones with more persistent and robust resistance.

In the pulsed treatment scenario, a dose of the ADC above its IC_50_ value is commonly used [25,31,34]. This treatment is given for some days and then is removed to allow surviving cells to recover. This cycling therapy protocol is conducted for several rounds, usually for months. This method is thought to be more clinically relevant because it is expected to simulate more closely the patient scenario.

Once resistant cells are generated, they can be pooled or individual clones selected. In the case of adherent cells, single colonies can be picked and grown individually. The isolation of single clones from non-adherent cells can be made by limiting dilution, seeding cells at a very low density, or by cytometric cell sorting.

An important aspect that requires consideration is the persistence of the resistant phenotype [25,51]. In fact, reversal of resistance may occur and to avoid the generation of confusing data while studying the mechanisms of resistance, frequent analyses of the stability of the resistant phenotype need to be done. Ideally, the resistance should be maintained over time in the absence of the drug. In general, the resistance obtained by high doses of the ADC is stable and does not require the continuous presence of the ADC in the culture media [50]. That is the most favorable situation, as maintaining pressure with the ADC may result in expensive and, more importantly, may impede the evaluation of certain experiments if the ADC is already present in the cultures of the resistant cells. In that case, the drug could be removed from the media 1–2 days before performing the experiments. Yet, under some circumstances, re-treatment of the resistant cells with the ADC should be considered. In fact, if periodic proliferation tests (recommended to ensure that the resistance to the ADC persists) indicate partial or complete loss of resistance, maintenance of cultures under ADC pressure is an adequate strategy, although it can reduce the proliferation of the resistant cells. To prevent reversion of the phenotype, periodic pressure (e.g. one month with and two months without) with the concentration of the ADC used to generate resistance can be done.

A recommended procedure to maintain a stock of resistant cells is to expand the cultures once the resistant cells are generated and to freeze several vials of the same batch. That is important to rescue the cells in case they lose resistance with time in culture. Additionally it is important to verify the resistant phenotype upon cycles of freezing and thawing [49].

##### Continuous Treatment

Cianfriglia et al. generated HL60 cells resistant to GO by maintaining the cell cultures in the presence of 5 ng/mL of GO for two months [52]. Of note, when they cultured the GO-resistant cell line for 2 weeks in GO-free media, they obtained revertant cells. Other authors also generated GO-resistant cells from HL60 cells but by continuous stepwise increasing doses of GO [28]. In that work, GO (100 ng/mL) was kept in the media of HL60/GO-resistant cells, replacing it every 3 days.

Lewis and colleagues created three lymphoma cell lines with acquired resistance to BV [32]. They continuously treated cells with increasing concentrations of BV over time. Chen et al. also generated different BV-resistant cell lines using constant or intermittent treatment [34]. Karpas-299 cells were treated at sub-IC_50_ concentrations (10 ng/mL) of BV for 1 month. Then cells were incrementally incubated with BV concentrations up to the IC_50_ value.

Li and colleagues generated several models of T-DM1 resistance. In that case, HER2+ cancer cell lines (BT474, NCI-N87, SKOV-3, and MDA-MB-361) were treated with gradually increasing concentrations of T-DM1 for 4–12 months [40]. In our group, we have also generated T-DM1 resistant clones from the BT474 HER2+ cell line [50]. The most practical approach was to seed BT474 cells at low density (5000 cells in 150 mm dishes) and treat them continuously with 5 nM T-DM1 for 3 months. T-DM1 resistant clones were isolated by ring cloning and maintained in culture media without the ADC [50]. Another protocol used by others to generate T-DM1-resistant BT474 cells was based on the use of increasing concentrations of T-DM1 (from 10 ng/mL to 1 μg/mL) for one year and then selecting clones through the limiting dilution method [53]. Guangmin et al. treated KPL-4 and BT-474M1 breast cancer cells by continuous treatment with increasing concentrations of T-DM1. The established T-DM1–resistant pools were maintained in culture with T-DM1 but two resistant cells lines, used for xenograft studies, were grown in T-DM1-free medium for 6 months [24]. In another study, BT474 and SKBR3 HER2+ breast cancer cell lines were cultured with constant increasing doses of T-DM1 (from IC_25_ to 4 μg/mL in BT474 and 0.05 μg/mL in SKBR3) over a 9-month period [39].

Other authors exposed NCI-N87, a HER2+ gastric cancer cell line, to gradually increasing concentrations (from 0.1 to 4 µg/mL) of T-DM1 for 6 months to generate resistance. Once established, the N87-TDMR resistant cell line was subcultured in a medium with 4 µg/mL T-DM1 [26]. Wang et al. also generated T-DM1 resistant cells from NCI-N87. In this case, cells were chronically exposed to progressively increasing concentrations of T-DM1 (from 50 ng/mL to 1 µg/mL). After 18 months, they selected a pool of T-DM1-resistant cells, and T-DM1-resistant clones were isolated by the limiting dilution method [54]. T-DM1–resistant HER2+ gastric cancer cell lines have also been generated by exposing OE-19 and NCI-N87 cells to increasing concentrations of T-DM1. Cells were initially treated with 0.12 µg/mL (N-87) or 0.08 µg/mL (OE-19) of the drug and then gradually increasing doses up to 2 µg/mL over 7 or 9 months [23]. Sauveur and colleagues generated T-DM1-resistant cell lines from esophageal OE-19 and breast MDA-MB-361 HER2+ cells [55,56]. The starting dose was 20% of the IC_50_ up to 0.3 nM (6 times the IC_50_) in the case of OE-19 and up to 0.4 nM (twice the IC_50_) in MDA-MB-361. The novelty of their studies relies on the fact that they cultured cells with T-DM1 in the absence or presence of the ABCB1/MDR1/P-gp modulator cyclosporin A [55,56]. The idea was to add cyclosporin A during the selection of T-DM1 resistant cells to prevent MDR1-mediated resistance.

Endo et al. treated JIMT-1 cells with increasing concentrations of T-DM1 from 0.2 to 2 μg/mL for 2 months. Then, cells were cultured with 3.6–4.0 μg/mL T-DM1 for 3 months and the resistant cells were always cultured in the presence of T-DM1 [45].

One in vitro resistant model was generated using a cell line obtained from a patient-derived xenograft (PDX). In this case, Nadal-Serrano et al. developed models of acquired resistance using the PDX118 cell line obtained from a mouse that was implanted with a cutaneous metastasis of a patient with HER2+ breast cancer. They treated cells chronically with T-DM1 for 45–90 days and established three resistant clones (R44, R55 y R200) [57].

##### Intermittent Treatment

Chen et al. generated a BV-resistant cell line using an intermittent treatment schedule in the L428 cell line [34]. Cells were incubated at a supra-IC_50_ concentration (50 µg/mL) until proliferation stopped. Then cells were allowed to recover in BV-free media. After recovery, BV was added back at the same dose. They finished the cycle scheme when constant growth was reported at supra-IC_50_ concentration of BV. Recently, Chen et al. also generated another BV–resistant Hodgkin lymphoma cell line model using the same pulse approach in the KMH2 cell line. The selection was achieved when constant proliferation was obtained at 20 µg/mL of BV [33]. Wei et al. also generated BV-resistant clones from KMH2 and L428 Hodgkin lymphoma lines [31]. Firstly, they produced BV-resistant pools by treating the cells at the IC_90_ dose of BV for one week and then allowed them to recover without the drug. This pulsed schedule was made three more times (for 7 weeks in total). Afterward, the pools were single-cell cloned in the presence of BV.

Loganzo et al. generated MDA-MB-361 and JIMT-1 HER2+ breast cancer cells lines resistant to TM-ADC (trastuzumab–maytansinoid ADC, structurally similar to T-DM1) by multiple cycles for 3–4 months of 3 days of approximately IC_80_ TM-ADC treatment, followed by 4–11 days of incubation of the cells in the drug-free medium [25]. Resistance to the drug was developed within 1.5–3 months and drug selection pressure was removed after approximately 3–4 months of these cyclic treatments. The same group also generated resistant cell lines to TM-ADC in other HER2+ cell lines (NCI-N87, HCC1954, and BT474) following a similar cyclic dosing schedule [51]. In addition, they generated clones from N87-TM cells using the technique of limiting dilution. Single cells were seeded in individual wells of a 96-well plate and treated with 100 nM TM-ADC. They reported that the resistant phenotypes remained stable for approximately 3–6 months [25,51].

Schwarz et al. also established T-DM1-resistant breast cancer cell lines by long-term cyclic treatment. They treated UACC89 and HCC1954 cells with cycles of 0.5 µg/mL T-DM1 for 3 days and 10 days off-treatment. They did several cycles for more than 6 months [48]. Other authors used HER2+ breast cancer cell lines (HCC1954, HCC1419, SKBR3, and BT474) to generate T-DM1 resistance but they were not able to develop BT474 resistant cells [58]. They generated resistant cells lines from the other models in 54 days by 3 consecutive cycles of 3 days on treatment followed by 3 days without the drug. This pulsed treatment was made for each T-DM1 concentration (1, 2, and 4 µg/mL). Recently, HCC1954 cells resistant to T-DM1 have been generated by treating cells with T-DM1 for 4 days, followed by 7 days in T-DM1-free media. In each cycle, the T-DM1 dose was increased from 20 ng/mL until ≥2 μg/mL [27].

In conclusion, both methods have successfully generated several models of secondary ADC resistance. As we have commented before, in the case of primary resistance, the main identified cause of resistance is the overexpression of ABC transporters. Such a mechanism has also been identified as a cause of secondary resistance. In addition, other mechanisms of resistance have also been described such as (1) reduction in target protein expression, (2) lysosomal proteolytic alterations, (3) disfunction of transporters that export the payload, (4) ADC trafficking defect, and (5) deregulation in proteins involved in signaling pathways (Table 2).

#### 3.1.3. Resistance to ADCs in Primary Cultures of Human Tumoral Cells

Although immortalized cell lines have represented exceptional models in molecular oncology, including drug therapy and resistance studies, ex vivo cultures using tumoral cells from patients mimic better the diversity, heterogeneity, and drug-resistant phenotypes present in the clinic scenario [20]. For these reasons, primary ex vivo cultures are also used in the ADC-resistance field.

With the aim of describing the role of ABCB1/MDR1/P-gp and the MDR phenotype in clinical GO resistance, Linenberger and colleagues analyzed pretreatment blast cells obtained from relapsed AML patients eligible for phase II clinical trials with GO [59]. Haag et al. used primary cells from patients with AML to test their sensitivity to GO and obtained GO-resistant primary cells [35]. Amico et al. isolated peripheral blood mononuclear cells from patients with AML and cultured them in vitro with IL-3 and GM-CSF and in the presence of increasing concentrations of GO, obtaining GO-resistant cells [36]. Walter et al. also validated GO-refractory primary cultures from AML patients, describing that they usually express functional ABC transporters and the inhibition of those transporters could increase GO cytotoxicity [29,30]. Rosen et al. also studied GO-resistance in blasts from patients with newly diagnosed AML [37]. Recently Islam et al. used cells isolated from HER2+ breast cancer patients’ samples to study the role of ROR1 in T-DM1 resistance [60].

### 3.2. In Vivo Models

In vivo models mimic more reliably the physiological conditions, including the variability of cellular components present in the tumor microenvironment. Indeed, the in vivo setting represents a useful method to evaluate the pharmacology, pharmacokinetics, efficacy, or safety of a potential therapy.

The most frequently used in vivo models are based on mice xenografted with either cell lines or patient-derived tumor tissue (patient-derived xenografts or PDX) (Figure 3) [61,62]. Cell line-derived xenografts are established by implanting tumoral cell lines in immunodeficient mice. Although these models have been broadly used, they have some inconveniences. One of them is related to the molecular alterations that accumulate tumoral cells upon long-term in vitro culture. Another element that is missing in the tumors created by implanting cancer cells is the lack of human stromal elements, which are present in human tumors and may contribute to the oncogenic characteristics of those tumors [63]. Due to these shortcomings, the use of PDXs has emerged as a more physiological alternative. The PDXs are generated by the implantation of tumoral tissue from a patient into immunocompromised mice [64]. Although is a more novel method, PDXs present some inconveniences, such as availability, low engraftment rate, or the long time usually required to be grown.

#### 3.2.1. Cell Line Derived Xenografts (CDXs)

##### Implantation of ADC Resistant Cells

ADC-resistant cell lines generated in vitro can be implanted into immunocompromised mice to verify whether their resistance is also present in the in vivo setting. This approach has been successfully used by several researchers [23,25,26,27,33,40,43,48,50,51,53,54]. In the case of Li and colleagues, the resistant phenotype reported in the T-DM1–resistant NCI-N87 cell line in vitro was not translated into resistance in the in vivo setting [40]. To establish a resistance model in vivo, large refractory tumors were dissected and transplanted in new mice that were treated with T-DM1 at a 3 mg/kg dose until the tumors were refractory. Failure to translate the in vitro resistant phenotype into the in vivo resistance has also been observed by us, especially in the case of the nude antibody trastuzumab. A pool of BT474 cells resistant to trastuzumab in vitro (BTRH) was still sensitive to that antibody in vivo. However, a clone resistant to trastuzumab in vitro (BTRH#10) was also resistant in vivo [65]. The reason for such discrepancy is at present unknown but has also been observed by others [40,66]. In our own experience in vitro resistance models do not always maintain their resistance when injected in mice suggesting that the interaction of resistant cells within their environment can modulate the efficacy of the drug. However, it is unclear if in vivo models can recapitulate better the resistance observed in patients compared with in vitro ones so for the time being all models are appropriate.

Another example of CDX was generated by D’Amico et al. Firstly they generated a murine breast cancer cell line overexpressing human HER2 that was resistant to trastuzumab and T-DM1. When they implanted these cells into Balb/c mice, tumors were also resistant to both anti-HER2 therapies [67].

##### Generation of ADC Resistance In Vivo

Resistance to ADCs can be modeled in vivo. Indeed, one of the first models of ADC resistance was generated using that approach [66]. Starling et al. implanted UCLA-P3 cells in nude mice and treated them with a vinca-conjugated anti-KS1/4 ADC. After cessation of therapy, tumor growth is usually reinitiated. One resistance was generated, the tumoral cells were isolated, grown in culture, and reimplanted into mice to generate new resistant variants. A remarkable observation was that cells retained resistance in vivo but not in vitro. Therefore, the in vivo resistance to an ADC does not necessarily translate into in vitro resistance and vice versa, as described before. This conclusion is in line with our own experience with an MMAF-based ADC targeting HER3 [65]. Studies in mice injected with breast cancer cells expressing HER3 and treated with the MMAF-anti HER3 ADC resulted in the generation of in vivo resistant tumors in a few of them. When cells from these tumors were placed in culture, they resulted sensitive to the antiproliferative action of the ADC (unpublished data). Therefore in vivo resistance to an ADC does not necessarily mean cell-autonomous resistance by the tumoral cells. In fact, that circumstance may be due to the stromal components present in the in vivo setting, which may promote a resistant phenotype.

Yu et al. established in vivo resistant tumors to an anti-CD22-vc-MMAE ADC [68]. They subcutaneously injected BJAB-luc and WSU-DLCL2 non-Hodgkin lymphoma cell lines in immunodeficient mice. When tumors reached 200 to 500 mm^3^, animals were treated with anti-CD22-vc-MMAE. When tumors regrew, mice were rechallenged again with gradually increasing doses of the ADC until obtaining refractory tumors. Once resistance was achieved, tumors were isolated and grown in culture to generate resistant cell lines. The generated cell lines were resistant to two ADCs (anti-CD22-vc-MMAE and anti-CD79b-vc-MMAE) not only in vitro but also in vivo. In this case, resistance was attributed to overexpression of ABCB1/MDR1/P-gp.

Recently, Graziani et al. generated PT-DM1 (trastuzumab-maytansinoid conjugate) resistant tumors after subcutaneous implantation of NCI-N87 cells in nude mice. When tumors reached approximately 300 mm^3^, mice were treated weekly with PBS or PT-DM1 (6 mg/kg) for 30 weeks. Most tumors responded to PT-DM1 treatment with a reduction in tumor volume. However, when some individual tumors reached a volume of 500 mm^3^ they were considered resistant [69]. Lirie and colleagues also generated an in vivo T-DM1 resistant model by implantation of NCI-N87 cells in nude mice and treatment of those mice with T-DM1 (10 mg/kg every 3 weeks) [70]. The initial refractory tumors were implanted in new mice three times to continue the treatment schedule and to increase the resistant phenotype. Recently, a new model of HER2 heterogeneity and T-DM1 resistance has been created by mixing HER2+ JIMT-1 cells and HER2 negative MDA-MB-231 cells in a 4:1 ratio before implantation in mice [27].

#### 3.2.2. Patient-Derived Xenografts (PDXs)

Several studies have reported the generation of models of resistance to ADCs by using PDXs. Ogitani et al. established three PDX models unresponsive to T-DM1. Two of them expressed low levels of HER2, which may explain the lack of response. However, one of them with high HER2 expression showed resistance to T-DM1 [44]. Kinneer and colleagues established a PDX model (ST1616B) from a lung metastasis refractory to T-DM1. Of note, the model responded to T-DM1 in vivo. For this reason, they generated a resistant variant (ST1616B/TDR) by treating the previous PDX version with T-DM1 over three passages [71]. Skidmore et al. obtained two PDX models (T226 and MAX1162F) derived from HER2+ breast cancer and confirmed T-DM1 resistance to a single dose of 3 mg/kg [42]. Recently, Li et al. described one lung PDX model bearing *ERBB2* amplification and mutation (S310F) that initially responded to T-DM1 but developed resistance over time [72]. Nadal-Serrano and colleagues established HER2+ PDXs resistant to T-DM1 in vivo, obtained by treating two sensitive PDX models (PDX118 or PDX510) with T-DM1 (15 mg/kg every 3 weeks) [57]. In addition, one PDX model (PDX580) was primarily resistant to T-DM1 in vivo because it was generated from a HER2+ patient tumor that progressed on T-DM1 therapy. Recently, Irie et al. established a PDX model from a HER2+ breast cancer insensitive to trastuzumab, pertuzumab, T-DM1, and chemotherapies [70].

An interesting but still poorly exploited model for the study of ADC resistance is represented by patient-derived organoids (PDOs). In this model, the patient’s tumor is mechanically and/or enzymatically treated, and then, the cells are incorporated into Matrigel to generate a 3D model. The PDOs can be maintained in culture and/or can be injected into immunocompromised mice [73]. These models are gaining presence in molecular oncology studies, particularly in the case of solid tumors, and may constitute an ex vivo 3D model closer to the tumor environment. Indeed, PDOs retain tumoral features such as cell–cell and cell–stroma interactions, intertumor heterogeneity, tumoral microenvironment, and therapeutic resistance [74]. Studies on those models are expected to yield important information about the efficacy of ADCs and their bystander effect, especially in heterogeneous tumors.

## 4. Conclusions and Perspectives

ADCs are emerging as a new family of therapies with excellent clinical opportunities due to the combination of a high grade of specificity (antibody) and a potent antitumor activity (payload). As for most of the therapies currently available, resistance to ADCs is a clinical problem. To fight that, better knowledge of the mechanisms of action and resistance to the ADCs are required. With respect to the latter, several models have been used to gain insights into those mechanisms and how to fight resistance.

Each preclinical model of ADC resistance has its own benefits and limitations. Thus, cell lines are cheap, unlimited, and their manipulation is easy. However, they usually bear a higher number of molecular alterations than primary tumoral cells. The latter represents a model closer to the clinical situation, but also offers difficulties such as availability or restricted growth ex vivo. On the other hand, in vivo xenograft models created by the injection of human tumoral cell lines in immunosuppressed mice allow the analysis of the efficacy of the drug in a physiological setting and can also inform about specific and non-specific toxicities of the ADC. In addition, these models may better mimic the clinical situation and they also allow the analysis of the toxicity and metabolism of the ADC and represent the complexity of the tumoral microenvironment. However, the in vivo models have some disadvantages such as the requirement of specific facilities, are more expensive, and do not exactly represent the stromal patient conditions for which PDXs may be closer to the human scenario. In general, in vitro models with immortalized human cell lines and ex vivo primary cultures are used to identify mechanisms of resistance. On the other hand, in vivo models are implemented to try novel therapies against ADC-resistant models. Some PDX models are used to validate in vitro discoveries about ADC resistance mechanisms.

The use of the preclinical models described above have allowed us to gain insights into the mechanisms of action of the ADCs and are also critical to uncover the mechanisms responsible for ADC resistance. These advances in the knowledge of ADC physiology, together with innovations in the technical development of ADCs, such as higher drug-to-antibody ratios, antibodies loaded with two or more cytotoxics, novel linkers, or bispecific antibodies are expected to improve the therapeutic possibilities of the ADCs. For instance, switching and replacing the components of the ADC has resulted in the generation of novel ADCs that overcome acquired resistance. That is exemplified by trastuzumab-deruxtecan, used in patients who become refractory to T-DM1.

## Figures and Tables

**Figure 1 cancers-13-04631-f001:**
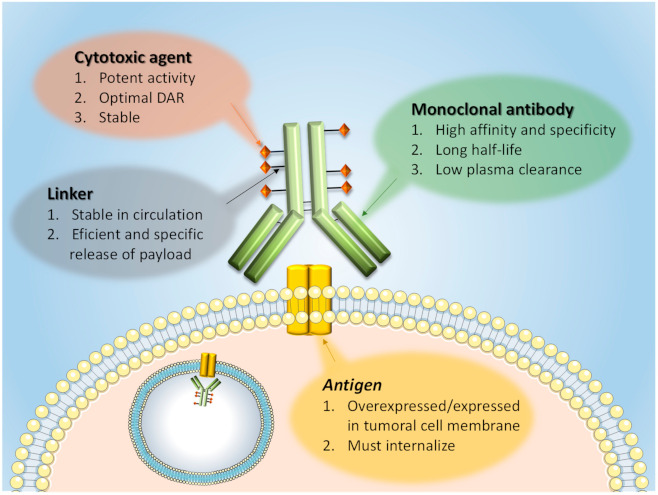
ADC general structure. Some important characteristics of the main ADC components are shown. The basic characteristics of the membrane antigens against which the ADCs are developed are also indicated. DAR = drug-to-antibody ratio.

**Figure 2 cancers-13-04631-f002:**
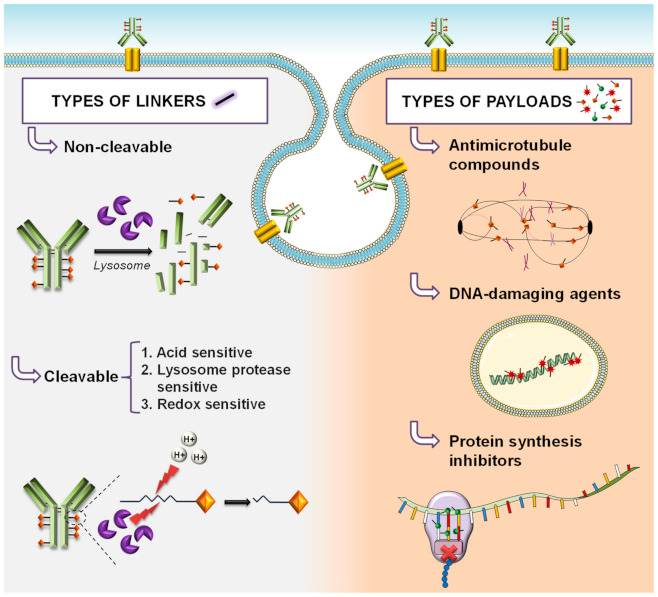
Types of linkers and payloads in ADC development.

**Figure 3 cancers-13-04631-f003:**
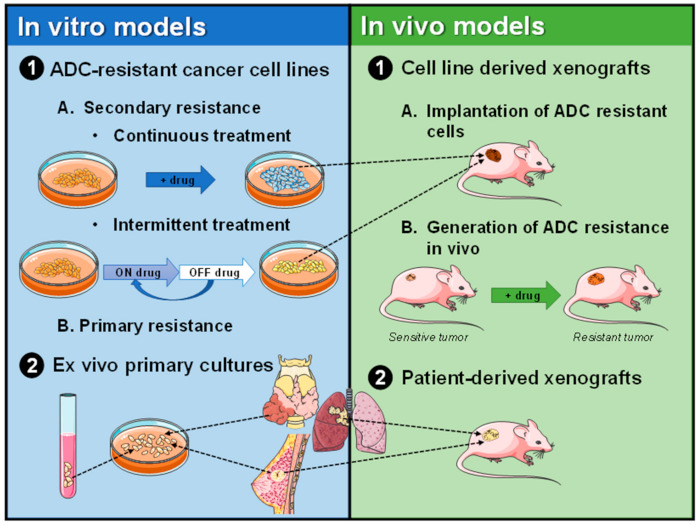
Schematic representation of the generation of preclinical models of ADC resistance.

**Table 1 cancers-13-04631-t001:** ADCs approved by the FDA.

ADC (Other Names, Commercial Name)	Target (Isotype Immunoglobulin)	Payload (Mechanism of Action)	Linker	Indication (Year of FDA Approval)	Company
Gemtuzumab ozogamicin (GO, Mylotarg^®^)	CD33 (IgG4)	Calicheamicin γ1 (CAL, cytotoxic antibiotic, DNA damage)	Cleavable hydrazone linker	Acute myeloid leukemia (AML) (2000–2010, 2017)	Pfizer (New York, U.S.)
Brentuximab vedotin (BV, SGN-35, Adcetris^®^)	CD30 (cAC10 chimeric IgG1)	Monomethyl auristatin E (MMAE, inhibition of microtubule polymerization)	Protease-cleavable linker, maleimidocaproyl-valine-citrulline-p-aminobenzyloxycarbonyl linker	Relapsed/refractory Hodgkin lymphoma and anaplastic large cell lymphoma (2011)	Seattle Genetics/Takeda(Washington, U.S./Tokyo, Japan)
Ado-trastuzumab emtansine (T-DM1, Kadcyla^®^)	HER2 (IgG1)	Mertansine (DM1, inhibition of microtubule polymerization)	Non-cleavable thioether linker	HER2+ metastatic breast cancer (2013) Adjuvant treatment of patients with HER2+ early breast cancer who have residual invasive disease after neoadjuvant taxane and trastuzumab based treatment (2019)	Roche (Basel, Switzerland)
Inotuzumab ozogamicin (INO, CMC-544, Besponsa^®^)	CD22 (IgG4)	Calicheamicin derivative (CAL, cytotoxic antibiotic, DNA damage)	Cleavable hydrazone linker	Acute lymphocytic leukemia (ALL) and chronic lymphocytic leukemia (CLL) (2017)	Pfizer (New York, U.S.)
Moxetumomab pasudotox (CAT-8015, Lumoxiti^®^)	CD22 (Fv portion of the mAb)	PE38, a 38 kDa fragment of Pseudomonas exotoxin A (inhibition of protein synthesis)	Payload fused to the antibody using a cleavable amino acid linker	Relapsed or refractory hairy cell leukemia (HCL) (2018)	AstraZeneca (Cambridge, UK)
Trastuzumab deruxtecan (T-DXd, DS-8201a, Enhertu^®^)	HER2 (IgG1)	Deruxtecan (DXd, topoisomerase I targeting)	Enzymatically cleavable maleimide glycynglycynphenylalanyn-glycyn peptide linker	HER2+ metastatic breast cancer (2019) HER2+ locally advanced or metastatic gastric cancer (2021)	AstraZeneca/Daiichi Sankyo (Cambridge, UK/ Tokyo, Japan)
Polatuzumab vedotin-piiq (PV, DCDS4501A, RG7596, Polivy^®^)	CD79b (IgG1)	Monomethyl auristatin E (MMAE, inhibition of microtubule polymerization)	Protease-cleavable maleimidocaproyl-valine-citrulline-p-aminobenzoyloxycarbonyl linker	Diffuse large B-cell lymphoma (2019)	Roche (Basel, Switzerland)
Enfortumab vedotin (EV, ASG-22ME, AGS-22M6E, Padcev^®^)	Nectin 4 (IgG1)	Monomethyl auristatin E (MMAE, inhibition of microtubule polymerization)	Cleavable maleimidocaproylvaline-citrulline linker	Locally advanced or metastatic urothelial cancer (2019)	Seattle Genetics/Astellas (Washington, U.S./Tokyo, Japan)
Sacituzumab govitecan (SG, IMMU-132, Trodelby^®^)	TROP-2 (IgG1)	SN-38 (topoisomerase I targeting)	Cleavable ydrolysable Ph sensitive CL2A linker	Metastatic Triple Negative Breast Cancer (2020)	Immunomedics (New Jersey, U.S.)
Belantamab mafodotin (GSK2857916, Blenrep^®^)	BCMA, CD269 (IgG1)	Monomethyl auristatin F (MMAF, inhibition of microtubule polymerization)	Non-cleavable maleimidocaproyl linker	Relapsed and refractory multiple myeloma (2020)	GlaxoSmithKline (Brentford, UK)
Loncastuximab tesirine (ADCT-402, Zynlonta^®^)	CD19	Pyrrolobenzodiazepine (PBD) dimer toxin or SG3199 (DNA damage)	Valine-alanine cleavable, maleimide type linker	Relapsed or refractory large B-cell lymphoma (2021)	ADC Therapeutics S.A. (Lausanne, Switzerland)

**Table 2 cancers-13-04631-t002:** Models of secondary resistance to ADCs.

Continuous Treatment Strategy					
ADC	Cell Line	Time (Months)	Resistance Mechanism	In Vivo Test	Reference Year of Publication
GO	HL-60	2	Reduced CD33 expression	Not reported	[52] 2010
GO	HL-60	6	Increased ABCB1/MDR1/P-gp transporter	Not reported	[28] 2012
BV	DEL Karpas-299 L540cy	Not reported	ABCB1/MDR1/P-gp induction and/or loss of CD30	Not reported	[32] 2014
BV	Karpas-299	3	Reduced CD30 expression	Not reported	[34] 2015
T-DM1	BT474 NCI-N87 SKOV-3 MDA-MB-361	4–12	Not described	NCI-N87 resistant cells were also resistant in vivo after several passages in mice treated with T-DM1	[40] 2016
T-DM1	BT474	3	Altered lysosomal pH and decreased lysosomal proteolytic activity	Resistance in vivo	[50] 2017
T-DM1	NCI-N87	6	ABCC2 and ABCG2 upregulation	Resistance in vivo	[26] 2017
T-DM1	NCI-N87	18	Decreased lysosomal V-ATPase and lysine-MCC-DM1 production	Resistance in vivo	[54] 2017
T-DM1	KPL-4	10	Decreased HER2 and upregulation of ABCB1/MDR1/P-gp	Lack of resistance	[24] 2018
	BT-474M1		Loss of SLC46A3 and PTEN deficiency	Poor growth	
T-DM1	BT474 SKBR3	9	PLK1 upregulation	Not reported	[39] 2018
T-DM1	MDA-MB-361	6	Increased baseline aneuploidy and altered intracellular DM1 trafficking	Not reported	[55] 2020
T-DM1	OE-19	6	Changes in cell adhesion and the prostaglandin pathway	Not reported	[56] 2018
T-DM1	BT474	12	STAT3 activation	Resistance in vivo	[53] 2018
T-DM1	JIMT-1	5	Loss of HER2 and increase of EGFR expression	Not reported	[45] 2018
T-DM1	OE-19 NCI-N87	7–9	Slightly decreased expression of HER2, overexpression of ABCC1, ABCC2, and ABCG2, and changes in the disposal of T-DM1 on the secreted extracellular vesicles	RN-87 is resistant in vivo	[23] 2019
T-DM1	PDX118	1.5–3	Decreased HER2 levels, impairment of lysosomal function, or increased drug efflux	Not reported. Generation of other models of in vivo resistance (Section 3.2.2)	[57] 2020
**Pulsed-Selection Strategy**					
**ADC**	**Cell line**	**Schedule**	**Resistance mechanism**	**In vivo test**	**Reference** **Year of publication**
BV	L428	Treatment with 50 µg/mL of BV until no proliferation was observed. Then, when cells were recovered in free-BV media, BV was added	Increased ABCB1/MDR1/P-gp	Not reported	[34] 2015
BV	KMH2	Same approach as before	Upregulation ABCB1/MDR1/P-gp	Resistance in vivo	[33] 2020
Anti-HER2 trastuzumab–maytansinoid ADC (TM-ADC), which is structurally similar to T-DM1	MDA-MB-361	Multiple cycles for 1.5 to 3 months of 3 days with approximately IC_80_ of TM-ADC followed by 4–11 days without drug	Increased ABCC1/MRP1 transporter	Resistance in vivo	[25] 2015
	JIMT-1		Decreased HER2 levels	Not reported	
TM-ADC	HCC1954 BT474	Five cycles at 10 nM of TM-ADC for 3 days, followed by 4–11 days of recovery. Then, the cells were exposed to six cycles extra of 100 nM. Time taken approximately of 4 months	Decreased HER2	Not reported	[51] 2016
	NCI-N87		Trafficking defect, caveolae-mediated endocytosis of T-DM1	Resistance in vivo	
T-DM1	HCC1954 HCC1419 SKBR3	Three cycles of 3 days on and 3 days off treatment at 1, 2, and 4 µg/mL each.54 days in total.	Defective cyclin B1 induction by T-DM1	Not reported	[58] 2017
T-DM1	UACC893 HCC1954	3 days on/10 days off for more than 6 months	HCC1954-TDR expressed very low HER2 levels	HCC1954-TDR has also resistance in vivo	[48] 2017
BV	KMH2 L428	One week at IC_90_, one week without treatment over 7 weeks	Upregulation of NF-KBsignature genes mediated increasing ABCB1/MDR1/P-gp expression	Not reported	[31] 2020
T-DM1	HCC1954	T-DM1 for 4 days, followed by 7 days off treatment. Total time was 8 months	Not described. Attenuated HER2 expression	HCC1954-TDR has also resistance in vivo	[27] 2021

BV = Brentuximab vedotin, GO = Gemtuzumab ozogamicin, T-DM1 = trastuzumab-DM1.

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
