# Peer review of "Generation of Antibody-Drug Conjugate Resistant Models"

_cancers, 2021, doi:10.3390/cancers13184631_

Round 1

Reviewer 1 Report

In this review "Modelling resistance to antibody drug conjugates" of Gandullo-Sánchez et al., the authors condensate the current activities in the field of tumor model use for testing response to ADCs and unraveling developing resistance mechansism. In principle, the topic is of interest and especially the use of different types of tumor models for analyzing resistance mechanisms is currently even a "hot" topic. However, there are several points which have to be carefully adressed, before this manuscript could be accepted for publication:

  1. Please chortly introduce Pgp/MRP1 in 3.1.1 - moreover, it would be more logic and stringent to use MDR together with MRP all over the manuscript.
  2. Both paragraphs A) continous treatment and B) intermittent treatment are too wordy and somehow, the take-home message is missing.
  3. Line 275: if cells were selected in vitro, this was not a PDX.
  4. Line 206: 4 x [35]
  5. Overall, the dosing is too detailed descibed. This is not exactly what readers expect from a review. Mechanisms and explanations, even hypotheses are missing - just MDR/MRP is outlined a little bit. How could additional mechanisms be unraveled by taking advantage of the different model types?
  6. Speaking of model types: a critical analysis of the cons of cell line and especially CDX-derived results is clearly necessary. On the other side: what about using PDX (mentioned occassionally) and also PDO - here especially for secondary resistance analysis?

Author Response

Reviewer 1

In this review "Modelling resistance to antibody drug conjugates" of Gandullo-Sánchez et al., the authors condensate the current activities in the field of tumor model use for testing response to ADCs and unraveling developing resistance mechansism. In principle, the topic is of interest and especially the use of different types of tumor models for analyzing resistance mechanisms is currently even a "hot" topic. However, there are several points which have to be carefully adressed, before this manuscript could be accepted for publication:

1. Please chortly introduce Pgp/MRP1 in 3.1.1 - moreover, it would be more logic and stringent to use MDR together with MRP all over the manuscript.

Author’s response: Following the recommendation of this Reviewer, a paragraph with comments on the relevance of multidrug transporters in drug resistance has now been placed at the beginning of section 3.1.1.

2. Both paragraphs A) continous treatment and B) intermittent treatment are too wordy and somehow, the take-home message is missing.

Author’s response: The originality of this review is its description of the preclinical models of resistance to ADCs. We therefore feel that such description must be sufficiently exhaustive.

3. Line 275: if cells were selected in vitro, this was not a PDX.

Author’s response: The Reviewer is right. The resistance was raised by in vitro culturing cells derived from a PDX. The fact that resistance was raised in vitro is mentioned in the text, and that the cells originally derive from a PDX is also clearly stated, in our opinion. Yet, we have slightly modified the text to make it more understandable.  

4. Line 206: 4 x [35]

Author’s response: We have noticed that the submitted version had a repetition of the reference 35, four times. We have corrected it and thank the Reviewer for letting us know.

5. Overall, the dosing is too detailed descibed. This is not exactly what readers expect from a review. Mechanisms and explanations, even hypotheses are missing - just MDR/MRP is outlined a little bit. How could additional mechanisms be unraveled by taking advantage of the different model types?

Author’s response: Again, we wish to emphasize that the focus of this review is its description of the preclinical models of resistance to ADCs. This requires detailed description of the models and how they were created. On the other hand, description of the mechanisms of resistance have been covered by other reviews and this is also a reason why we did not enter into their detailed description. However, please note that we have included the described mechanisms of resistance in Table 2.

Obviously, the different models of resistance to different drugs may uncover novel mechanisms of resistance. That is one of the advantages of the research that can be done with those preclinical models. We do not feel that we can anticipate those potential novel mechanisms and therefore we prefer to be cautious on that. According to the point on MDR/MRP, we also discussed about the use of cyclosporin A in the development of ADC resistant models. The authors used this approach to avoid the generation of MDR/MRP alterations and study others possible mechanisms of resistance. We cannot relate one model of generation of resistance with a type of mechanism of resistance because any of them could generate different resistance by different mechanisms.

6. Speaking of model types: a critical analysis of the cons of cell line and especially CDX-derived results is clearly necessary. On the other side: what about using PDX (mentioned occassionally) and also PDO - here especially for secondary resistance analysis?

Author’s response: We agree with the Reviewer that some comments about the cons of using cell lines should be included in this review. In fact, they were included in the initial version of the paper. The  following paragraphs that referred to that are included also in  the revised version):

  • Paragraph 115-124.
  • Paragraph 335-3.
  • Paragraph 356-360.
  • Paragraph 468-4780.

The PDX models were outlined in the paper. In the case of PDO, we have not found much information about their use to generate models of ADC resistance. Yet, we consider PDOs may be useful to generate more physiological models of resistance and therefore, and following the indication of the Reviewer, we have added a paragraph on that in the revised version of the paper.

Reviewer 2 Report

In this review, the authors present model systems that can be used to study the effect of antibody-drug conjugates (ADCs) and the development of resistance. They give a comprehensive overview of this still manageable topic and present both in vitro and in vivo models. To be blunt, the only aspect that is not adequately addressed is the problem of undesirable side effects. In the introduction, ADCs are introduced and visualized in Figure 1. The typical linkers and modes of action are shown in Figure 2, before Table 1 lists the FDA-approved ADCs. This is followed by the models for ADC resistance. A distinction is made between primary and secondary resistance. Furthermore, the therapy regime can be intermittent or continuous. Both are mainly studied with cell lines, summarized for secondary resistance in Table 2; the listing of primary human cell models is much shorter. For in vivo models, those with PDX are much closer to human conditions. The authors conclude that multiple models are useful for overcoming secondary resistance in particular. 

Author Response

Reviewer 2

In this review, the authors present model systems that can be used to study the effect of antibody-drug conjugates (ADCs) and the development of resistance. They give a comprehensive overview of this still manageable topic and present both in vitro and in vivo models. To be blunt, the only aspect that is not adequately addressed is the problem of undesirable side effects. In the introduction, ADCs are introduced and visualized in Figure 1. The typical linkers and modes of action are shown in Figure 2, before Table 1 lists the FDA-approved ADCs. This is followed by the models for ADC resistance. A distinction is made between primary and secondary resistance. Furthermore, the therapy regime can be intermittent or continuous. Both are mainly studied with cell lines, summarized for secondary resistance in Table 2; the listing of primary human cell models is much shorter. For in vivo models, those with PDX are much closer to human conditions. The authors conclude that multiple models are useful for overcoming secondary resistance in particular. 

Author’s response: First of all, we wish to thank the Reviewer for the positive comments about the paper. Most of those comments refer to the content of the paper. The only criticism of the Reviewer refers to the lack of information about the side effects of the ADCs. While we feel that this is also an interesting topic, we consider that it falls beyond the scope of the paper we prepared. We will consider this idea for another potential review on those side effects.

Reviewer 3 Report

The paper "modelling resistance to antibody drug conjugates" was an interesting manuscript to read.  While I appreciated the details that the authors went into about cell culture conditions used to create the various resistant cell lines, I am greatly concerned that there was little or no discussion about what was learned from the various studies.  I came away from the article with a lot of knowledge about how to generate an ADC-resistant cell line, but virtually no knowledge of what drives ADC resistance and how these cell lines have helped to uncover mechanisms of ADC resistance.  In this respect, a better title for the article would be something like: "Generation of ADC-resistant cell lines and tumor models".   If this were the title of the article, I would be more comfortable with it.  But the current title gives the reader the impression that they are going to learn about ADC-resistance.  In fact, that isn't true.  They only learn about how to make ADC-resistant cells.
Where there were lots of references, there were also significant chunks without any references.  For example: Section 3.1.2 has only 1 reference. (was all the info taken from this one reference?  If so, maybe the author should shorten this section and say something like "Please see reference 44 for a more in-depth discussion of secondary resistance").  Similarly, the last paragraph of 3.2.2 describes PDOs, but has no references cited.  There were also significant language issues - especially in the 2nd half of the paper.

Author Response

The paper "modelling resistance to antibody drug conjugates" was an interesting manuscript to read.  While I appreciated the details that the authors went into about cell culture conditions used to create the various resistant cell lines, I am greatly concerned that there was little or no discussion about what was learned from the various studies.  I came away from the article with a lot of knowledge about how to generate an ADC-resistant cell line, but virtually no knowledge of what drives ADC resistance and how these cell lines have helped to uncover mechanisms of ADC resistance. 

Author’s response: Thanks for your comments. Indeed, the aim of this review is that of teaching and giving advice about how to generate ADC resistant models. There are several reviews discussing mechanisms of resistance, but no review has been published exhaustively covering how to develop the models. For that reason, our review focuses on that. Yet, in Table 2 we summarize the mechanisms of resistance discovered in the different in vitro resistant models discussed along the review. In addition, we have included a conclusion about mechanisms of resistance to ADCs at the end of section 3.1.2, before that Table.

In this respect, a better title for the article would be something like: "Generation of ADC-resistant cell lines and tumor models". If this were the title of the article, I would be more comfortable with it.  But the current title gives the reader the impression that they are going to learn about ADC-resistance.  In fact, that isn't true.  They only learn about how to make ADC-resistant cells.

Author’s response: we have changed the title to: “Generation of antibody-drug conjugate resistant models”.

Where there were lots of references, there were also significant chunks without any references.  For example: Section 3.1.2 has only 1 reference. (was all the info taken from this one reference?  If so, maybe the author should shorten this section and say something like "Please see reference 44 for a more in-depth discussion of secondary resistance"). 

Author’s response: We have revised the whole paper and added several additional references in various paragraphs of the text. With respect to section 3.1.2, it is noteworthy to mention that some comments in those paragraphs refer to our experience in the generation of the mentioned models. Therefore, in addition to the reference provided, these comments derive from our unpublished data and we consider that such information is helpful to understand how we prepared the ADC resistant models described there.

Similarly, the last paragraph of 3.2.2 describes PDOs, but has no references cited. 

Author’s response: We have included more information and two references.

There were also significant language issues - especially in the 2nd half of the paper.

Author’s response: We have revised the English of the whole review and made some changes.

Round 2

Reviewer 1 Report

Overall, I can not support the acceptance of this review because of the mentioned major criticism which has not been addressed properly by the authors. The argument raised is basically, that a very detailed description of the concentrations used is necessary for the type of review the authors intended to deliver. Which is acceptable, but is in my opinion not novel and relevant enough to justify publication in a relatively high ranking journal like CANCERS. I would thus recommend to look for a suited journal in the range of 3 impact points - the potential readership is definitely very limited for such a very special review.

Author Response

 We are sorry that the Reviewer is not satisfied by the comments and amendments made in the former revised version of the manuscript to address the questions he/she raised. We have re-read the comments of this Reviewer and made even more changes in this version: we included a conclusion about mechanisms of resistance to ADCs at the end of the “intermittent” protocols for the generation of resistant models. Moreover, please note that a short mention to those mechanisms was already included in Table 2. We have also tried to be more clear in some paragraphs by eliminating some information and references related to ADC concentrations. In conclusion, we have done our best to satisfy the Reviewer’s requests and tried to make the paper even more attractive for publication in CANCERS.

Reviewer 3 Report

The authors appear to have adequately addressed my concerns.